# Prevalence and predictors of anemia among six-week-old infants in Kwale County, Kenya: A cross-sectional study

Joyce Mwakishalua[1]*, Simon Karanja[1], Raphael Lihana[2], Collins Okoyo[3,4], Nicole Stoffel[5], Michael Zimmermann[6]

1 School of Public Health, Jomo Kenyatta University of Agriculture and Technology, Nairobi, Kenya, 2 Centre for Virus Research, Kenya Medical Research Institute, Nairobi, Kenya, 3 Eastern and Southern Africa Centre of International Parasite Control, Kenya Medical Research Institute, Nairobi, Kenya, 4 Department of Epidemiology, Statistics and Informatics, Kenya Medical Research Institute, Nairobi, Kenya, 5 Laboratory of Nutrition and Metabolic Epigenetics, Department of Health Sciences and Technology, ETH Zurich, Zurich, Switzerland, 6 Medical Research Council Translational Immune Discovery Unit, Weatherall Institute of Molecular Medicine, John Radcliffe Hospital, University of Oxford, Oxford, United Kingdom

* stellawali507@gmail.com

**Data Availability Statement:** The dataset supporting the conclusion of this article is included as a supplementary file.

## Abstract

Anemia is a significant public health problem among children worldwide. The etiology of anemia is multifactorial but iron deficiency (ID) is the most common cause of anemia in low- and middle-income countries. ID and anemia in infancy can impair growth and cognitive development. The aim of this study was to determine the prevalence and predictors of anemia among six-week-old infants in Kwale County, Kenya. This cross-sectional study included 424 mother-infant pairs. Structured questionnaires were administered to the mothers to obtain information on socio-demographic variables, maternal characteristics and birth information. Anthropometric data was collected for each child. A heel prick was done to measure hemoglobin and zinc protoporphyrin concentration levels. Chi-square test, bivariate and multivariate regression analyses were done to determine factors associated with anemia. The prevalence of ID, anemia and IDA was 60.4% (95%CI: 55.9–65.2), 21.0% (95%CI: 17.5–25.2) and 15.8% (95%CI: 12.7–19.7) respectively. Bivariate analysis showed that the risk of anemia was significantly higher among male infants (odds ratio (OR) = 2.20 (95%CI: 1.33–3.63), p = 0.002), iron deficient infants (OR = 2.35 (95%CI: 1.39–3.99), p = 0.001) and infants from Msambweni Sub-County (OR = 2.80 (95%CI: 1.40–4.62), p<0.001). Multivariate analysis revealed that odds of anemia were significantly higher in infants born to mothers who did not use iron supplements during pregnancy (adjusted odds ratio (aOR) = 74.01 (95%CI: 2.45–2238.21), p = 0.013 and significantly lower in infants born to mothers with parity ≥ 4 (aOR = 0.05 ((95%CI: 0.00–0.77), p = 0.024). In six-week-old infants in rural Kenya, anemia prevalence was 21.0% with ID accounting for 75.3% of anemia cases. Given the physical and cognitive impairments associated with ID and anemia in early infancy, it may be prudent to re-evaluate the current Kenyan pediatric protocols to include anemia screening and potential treatment of infants less than 6-months of age.

**Funding:** The authors received no specific funding for this work.

**Competing interests:** The authors have declared that no competing interests exist.

## Introduction

Anemia continues to be a significant public health problem affecting over a third of the global population with a worldwide prevalence of 22.8% across all ages [1] causing 68 million disability adjusted life years (DALYS) [2]. The World Health Organization (WHO) estimates that 42% of children < 5 years of age and 40% of pregnant women worldwide are anemic [3]. Regional estimates show that Sub-Saharan Africa bears the highest burden of anemia with prevalence of 62% [3]. Children under five years have the highest burden of anemia, predominantly in low- and middle-income countries (LMIC) and iron deficiency (ID) is the most common cause of anemia in this age group, accounting for ≥60% of the cases of anemia [2]. A higher anemia prevalence of 78.4% was found in Ghanaian children under five years of age [4]. In Kenya, anemia is already common in infants < 1 year of age e.g., 70% of 6-month-old infants in south coastal Kenya are anemic [5].

Anemia of any etiology is associated with increased risks for child mortality and morbidity [6]. More specifically, iron deficiency anemia (IDA) in early infancy may result in irreversible impaired cognitive development and reduce growth [7]. Although the cause of immune deficits in malnutrition are poorly understood, ID may impair adaptive immunity and thereby reduce efficacy of vaccines in early infancy [8, 9]. As a result, IDA reduces the productivity of these infants as they grow through adolescence into adulthood due to impaired cognitive ability and reduced physical development [10]. Other risk factors associated with anemia in the developing countries include: thalassemia and hemoglobinopathies, hemolysis or blood loss due to parasitic infections especially in malaria endemic areas, and other nutritional deficiencies including folate and vitamin A, C and B12 deficiencies [11]. Moreover, postnatal factors such as frequent illnesses like pneumonia, early introduction of low-iron-content cereal-based complimentary foods, rapid postnatal growth rate, low birth weight as well as infectious diseases like malaria, increase the likelihood of infants developing ID early, irrespective of whether they were born at term or not [12].

In Kenya, the nationwide prevalence of anemia in children under one year is 39.5% [13]. A study among 6 to 9-month-old infants in Keiyo sub-County reported an anemia prevalence of 21.7% [14]. A study in Kwale County among infants aged 3–6 months found the prevalence of ID, anemia and IDA to be 69%,52% and 38% respectively [15]; but the study was small (n = 25) and non-representative.

During early life, there is a critical timeframe for implementing health interventions, like routine childhood interventions. If these interventions are delayed beyond this point, they may become ineffective and lead to resurgence of vaccine-preventable diseases. Additionally, delayed interventions could contribute to poor growth, cognitive impairment, and potentially irreversible health issues in the future [15, 16]. In addition, ID may also be a previously unrecognized contributor to the reduced efficacy of vaccines in Sub-Saharan Africa [9]. Hence, it is crucial to assess the impact of anemia in early infancy as a factor contributing to elevated rates of child mortality and morbidity in Sub-Saharan Africa.

Provision of iron supplements to infants and children 6–24 months of age is recommended in areas where anemia is a severe public health problem i.e., prevalence ≥40% [17]. However, infants below six months of age have been exempted from many recommendations targeting prevention and treatment of ID and anemia in highly prevalent anemia regions. This is because it has generally been assumed that infants during their first six months of life are protected from IDA. This assumption stems from the general belief that infants build sufficient iron stores in their bodies during fetal development, providing protection against IDA in the first six months after birth and that anemia is not of concern during early stages of life [18]. However, this may not be the case in sub-Saharan Africa because sufficient iron is passed from

the mother to the fetus only if the mother has adequate iron stores during pregnancy and most pregnant women in their third trimester in Africa are anemic and iron deficient [2]. Although delayed cord clamping has recently been proposed to prevent ID and the resulting anemia in infants, it has been difficult to implement in sub-Saharan Africa because of under-developed healthcare systems and busy maternity wards [19].

Despite broad knowledge on anemia and ID in children, researchers point out that there is much that remains unknown about this problem in early infancy. Previous studies on anemia in Kenya have been restricted to infants above six months of age [5, 13, 14]. Furthermore, previous studies on anemia in LMIC with high malaria prevalence, where younger children bear the greatest burden of anemia, have not taken into account geographical factors when examining the connections between these outcomes and ecological or geographic factors [20]. The aim of this study therefore, was to determine the prevalence of anemia and ID and associated predictors among six-week-old infants in Kwale County, Kenya.

## Materials and methods

### Ethics statement

The approval for the study was sought from the Swiss Federal Institute of Technology (ETH) and the JKUAT Institutional Ethics review Committee, approval number; JKU/IERC/02316/ 0052. Additional approval was provided by the Kwale County health authorities. Written participant consent was obtained from the mothers.

### Study design and setting

This was a cross-sectional study conducted in Msambweni County Referral Hospital and Kwale Sub-County Hospital in Kwale County. The study was nested within a randomized clinical trial (RCT) [21]. The primary objective of the RCT was to determine whether detecting and correcting ID and IDA using well absorbed iron improves efficacy of vaccines and to define the specific immune mechanisms underlying this effect.

Kwale County is located in south coast of Kenya at Coordinates: $4°10'S$ $39°27'E$) bordering the Republic of Tanzania to the South West, Indian Ocean to the East, Taita-Taveta County to the West, Kilifi County to the North and Mombasa County to the North East. It has an estimated population of 866,820 [22].

Msambweni County Referral Hospital is a Level 5 government health facility. It offers both inpatient and outpatient medical services and has a bed capacity of 155. Being a referral hospital, it receives several cases from different areas within the county and therefore a good representative of the county population.

Kwale Sub-County Hospital is a high-volume level 4 government health facility within Matuga Sub-County. It has a bed capacity of 62 and offers both inpatient and outpatient medical services.

### Sample size determination

Sample size was calculated using Fisher's formula [23]; $n = (z^2pq)/e^2$; where Z is the value from standard normal distribution corresponding to 95% confidence level (Z = 1.96), p is the estimated prevalence of anemia among infants in Kwale County (assumed as 50%), q is the proportion of infants without anemia (1-p), e is the margin of error assumed as 5%. The final sample size was thus calculated as 424 mother-infant pairs after factoring in a 10% non-response rate. Simple random sampling was used to select the final sample size from the total number of mother-infant pairs who were eligible to be enrolled in the study. The participants

were randomly sampled from the number of mother-infant pairs that came for screening for the RCT.

## Survey procedures

The study was carried out between January and April 2023. During recruitment, the study purpose was explained to the mothers and those who were willing to participate were scheduled for a screening date at the hospital when their child would be exactly 6 weeks old ± 3 days. On the day of screening, the study aim was explained again in detail to the mothers who signed consent forms both in English and Swahili and each mother-infant pair was given a unique identification number. A screening form was programmed onto redcap and administered using tablets. The study included HIV negative mothers above 15 years of age with infant aged 6 weeks ± 3 days, born via vaginal delivery, full-time breastfed, not having received any vaccine beyond the birth dose of oral polio vaccine (OPV) and Bacillus Calmette-Guerin (BCG), and with no medical condition that precluded study involvement. The study excluded infants with conditions that might have affected participation such as hematological and non-hematological malignancies, chronic kidney disease, diabetes and inability to provide maternal informed consent. Mother-infant pairs who met the inclusion criteria were enrolled into the study.

## Data collection

An interviewer administered questionnaire was used to collect data on socio-demographic and maternal characteristics. Date of birth, maternal age, hemoglobin (Hb) level during last antenatal care (ANC) visit and parity and other factors were subsequently confirmed from the mother-child ANC booklet or immunization card or exercise book. A standardized SECA 813 digital measurement scale was used to measure infant's body weight to the nearest 0.01kg. A standard measurement board from UNICEF was used to measure infants' length/height to the nearest 0.5cm. A heel prick was done using single use sterile lancet and Hb concentration measured on capillary blood using a Hemocue 301 system device (Hemocue, Sweden). ZnPP concentration was measured on capillary blood using iCheck Anemia device (Bioanalyt, Germany). The device uses photometric determination of ZnPP by auto-florescence. Both machines have quality controls (two levels; high and low) which were performed prior to measuring samples during screening.

## Data management and analysis

The WHO anthropometrics software was used to calculate infant Z-scores for weight for length (WLZ) which indicates wasting, weight for age (WAZ) which indicates underweight and length for age (LAZ) which indicates stunting. Both questionnaires and laboratory diary information were programmed into tablets then used to capture data electronically on Redcap (Vanderbilt University) which had inbuilt data quality checks to reduce errors during the data entry process. Collected data was sent to the server daily. Stata version 16 (StataCorp) was used for statistical analyses. Chi square test was used to show association between predictor variables and anemia status.

Both maternal and infant factors related to anemia were analyzed by bivariate analysis and strength of association reported as odds ratio (OR) with 95% confidence intervals (CI). Variable multicollinearity was assessed to identify variables that were strongly correlated. To select variables for multivariate analysis an inclusion criterion of p-value <0.5 was used in a stepwise variable selection method. A multivariate analysis was done using the selected variables and strength of association reported as adjusted OR (aOR) with 95% CI.

## Results

### Sociodemographic characteristics of participants

Overall, 424 mother-infant pairs were enrolled in the study; 211 from Matuga sub-County and 213 from Msambweni sub-County. Of the 211 participants from Matuga sub-County, 82 (19.3%) were from Mkongani location, 117(27.6%) from Tsimba Golini location and 12 (2.8%) from Waa location. Of the 213 participants from Msambweni sub-County, 13 (3.1%) were from Kinondo location, 74 (17.5%) from Ramisi location, 100 (23.6%) from Ukunda location, and 26 (6.1%) from Gombato location (Table 1).

All infants in the study were aged 6 weeks (± 3 days) old. 55% of the infants were male while 45% were female. The mean (SD) birth weight of the infants was 3.10 (0.5) kg. Mean (SD) weight of the infants was 4.7 (0.6) kg and mean (SD) height was 54.7 (2.1) cm. Mean (SD) head circumference was 37.9 (1.6) cm and mean (SD) gestational weeks at birth was 37 (1) weeks. Mean (SD) Hb concentration of the infants was 12.3 (1.5) g/dl. Mean (SD) ZnPP concentration of the infants was 44.0 (13.1) μmol/mol Hb.

Mean (SD) age of mothers in the study was 27.7 (6.1) years. Mean (SD) weight of the mothers was 59.8 (12.3) kg. The median number of children per mother was 3 with a minimum of 1 child and a maximum of 11 children. Mean (SD) Hb concentration of the mothers during the last ANC visit was 10.8 (1.4) g/dl. Maternal information on education level and use of iron supplements during pregnancy was available for 157 mothers of whom 98.1% reported to having used iron supplements during pregnancy while 2% reported to not have used iron supplements during pregnancy. 17.8% of mothers had attained a post-secondary level of education, 21.7% had attained a secondary education level, 54.1% had attained a primary education level and 6.4% had no education level.

### Nutritional outcomes in infants

The overall prevalence of ID, anemia and IDA among infants in Kwale county was 60.4% (95%CI: 55.9–65.2), 21.0% (95%CI: 17.5–25.2) and 15.8% (95%CI: 12.7–19.7) respectively. The prevalence of wasting, underweight and stunting was 2.1% (95%CI: 1.1–4.1), 0.2% (95%CI: 0.0–1.7) and 6.2% (95%CI: 4.3–9.0) respectively (Table 2).

Msambweni Sub-County had a higher prevalence of anemia 29.1% (95%CI:23.6–35.9) compared to Matuga Sub-County 12.8% (95%CI:9–18.2). Kinondo location in Msambweni Sub-County had the highest proportion of infants with anemia 50.4% (95%CI: 13.6–69.5) followed by Gombato Location 42.3% (95%CI:27.0–66.3) while Waa Location in Matuga Sub-County had no cases of anemia (Fig 1).

Male infants exhibited a significantly higher prevalence of anemia at 26.7% (95%CI:21.6–33.1) compared female infants, who had a prevalence of 14.2% (95%CI:10.0–20.2). Furthermore, infants with low birthweight had a significantly higher prevalence of anemia at 29% (95%CI:16.7–50.3) compared to infants with a birthweight of $\geq$ 2.5 kg, where the prevalence of anemia was 20.2% (95%CI:16.5–24.6). Infants weighing weighed < 5.0 kg had a lower anemia prevalence at 19.9% (95%CI: 15.8–25) compared to infants weighing $\geq$5.0 kg, with a prevalence 23.8% (95%CI:17.4–32.5). Similarly, infants with a head circumference <38.0 cm had a lower anemia prevalence at 20.3% (95%CI: 16.0–25.7) compared to infants with a head circumference of $\geq$38.0 cm, where the prevalence of anemia was 22.4% (95%CI:16.6–30.1). Based on gestational age at birth, infants born at <37 weeks gestation had a higher anemia prevalence at 24.5% (95%CI:15.3–39.3) compared to infants born at $\geq$ 37 weeks gestation, where the prevalence of anemia was 20.5% (95%CI:16.7–25.1). Additionally, based on infants' height, those with a height <55.0 cm had a higher prevalence of anemia at 22.0%

**Table 1. Socio-demographic and health information of the mothers and infants surveyed in Kwale County, Kenya.**

| Factor | Kwale (n = 211) | Msambweni (n = 213) | Overall (n = 424) |
|---|---|---|---|
| Location: | | | |
| Mkongani | 82 (19.3%) | - | 82 (19.3%) |
| Tsimba Golini | 117 (27.6%) | - | 117 (27.6%) |
| Waa | 12 (2.8%) | - | 12 (2.8%) |
| Kinondo | - | 13 (3.1%) | 13 (3.1%) |
| Ramisi | - | 74 (17.5%) | 74 (17.5%) |
| Ukunda | - | 100 (23.6%) | 100 (23.6%) |
| Gombato | - | 26 (6.1%) | 26 (6.1%) |
| Mother details: | | | |
| Age (in years); [mean (min-max)] | 27.6 (16–45) | 27.8 (18–47) | 27.7 (16–47) |
| Level of education | | | |
| None | 8 (10.5%) | 2 (2.5%) | 10 (6.4%) |
| Primary | 41 (54.0%) | 44 (54.3%) | 85 (54.1%) |
| Secondary | 12 (15.8%) | 22 (27.2%) | 34 (21.7%) |
| Post-secondary | 15 (19.7%) | 13 (16.1%) | 28 (17.8%) |
| Parity; [median (min-max)] | 3 (1–11) | 3 (1–8) | 3 (1–11) |
| Haemoglobin (g/dl); [mean (min-max) | 10.8(7.7–15.0) | 10.8 (7.0–14.0) | 10.8 (7.0–15.0) |
| Weight (in kg); [mean (min-max) | 58.2 (39.4–95.2) | 61.5 (35–112) | 59.5 (35–112) |
| Use iron supplement | | | |
| Yes | 75 (98.7%) | 79 (97.5%) | 154 (98.1%) |
| No | 1 (1.3%) | 2 (2.5%) | 3 (2%) |
| Number of ANC visits; [mean (min-max)] | 4 (2–8) | 5 (2–9) | 5 (2–9) |
| Had twin pregnancy | | | |
| Yes | 1 (1.3%) | 2 (2.5%) | 3 (2.0%) |
| No | 75 (98.7%) | 79 (97.5%) | 154 (98.1%) |
| Household water source | | | |
| Improved | 50 (65.8%) | 59 (72.3%) | 109 (69.4%) |
| Unimproved | 26 (34.2%) | 22 (27.2%) | 48 (30.6%) |
| Cost of travel to the nearest health facility (Ksh); [median (min-max)] | 126 (50–300) | 117 (50–500) | 122 (50–500) |
| Infant details: | | | |
| Gender | | | |
| Male | 108 (51.4%) | 124 (58.5%) | 232 (55%) |
| Female | 102 (48.6%) | 88 (41.5%) | 190 (45%) |
| Weight at birth (in kg); [mean (min-max)] | 3.1 (2.1–5.2) | 3.1 (2.0–4.9) | 3.10 (2–5.2) |
| Weight at 6 weeks (in kg); [mean (min-max)] | 4.6 (3.5–6.1) | 4.7 (3.4–6.5) | 4.7 (3.4–6.5) |
| Gestation (in weeks); [mean (min-max)] | 37.9 (34–40) | 37.7 (32–42) | 37.8 (32–42) |
| Height (in cm); [mean (min-max)] | 54.7 (48–60.3) | 54.7(48.5–61.4) | 54.7 (48–61.4) |
| Head circumference (in cm); [mean (min-max)] | 38.1 (35–49) | 37.8(35–54.4) | 37.9 (35–54.4) |
| Wasting | | | |
| Not wasted | 204 (96.7%) | 207 (99%) | 411 (97.9%) |
| Wasted | 7 (3.3%) | 2 (1%) | 9 (2.1%) |
| Underweight | | | |
| Not underweight | 210 (99.5%) | 209 (100%) | 419 (99.8%) |
| Underweight | 1 (0.5%) | 0 | 1 (0.2%) |
| Stunting | | | |
| Not stunted | 202 (95.7%) | 192 (91.9%) | 394 (93.8%) |
| Stunted | 9 (4.3%) | 17 (8.1%) | 26 (6.2%) |

*(Continued)*

**Table 1.** (Continued)

| Factor | Kwale (n = 211) | Msambweni (n = 213) | Overall (n = 424) |
|---|---|---|---|
| Haemoglobin; [mean (min-max) | 125.6 (95–162) | 118.6 (87–170) | 122.1 (87–170) |
| Zinc protoprophyrin; [mean (min-max) | 42.6 (19–125) | 45.4 (16–90) | 44 (16–125) |

(95%CI: 16.9–28.6) compared to infants with a height ≥55.0 kg, with a prevalence of 20.2% (95%CI:15.5–26.2).

Regarding mothers age, infants born to mothers aged 31–40 years had the highest prevalence of anemia at 24.4% (95%CI:17.8–33.4) and those born to mothers aged >40 years, had the least anemia prevalence at 8.3% (95%CI: 1.3–54.4). In terms of education level, infants of mothers with primary education had the highest prevalence of anemia at 22.4% (95%CI: 15.0–33.2) and infants of mothers with no formal education, had the lowest anemia prevalence at 10.0% (95%CI: 1.6–64.2). Infants born to mothers with parity one had the highest prevalence of anemia at 23.7% (95% CI: 16.4–34.1) and infants born of mothers with parity two, had the least prevalence of anemia at 18.8% (95%CI: 12.8–27.6). Concerning maternal anemia status during the last ANC visit, mothers who were anemic had a slightly higher proportion of infants with anemia 22.0% (95%CI: 16.6–29.1) compared to mothers who were not anemic during the last ANC visit, with a prevalence of 21.7% (95%CI: 16.2–19.1). The highest proportion of anemia in infants of 26.0% (95%CI:19.8–34.2) was in the category of mothers weighing 50–59 kgs while the least proportion of anemia in infants of 5.9% (95%CI: 1.5–22.6) was in the category of mothers weighing >80 kgs. Infants born of mothers who reported to have used iron supplements during pregnancy had an anemia prevalence of 18.2% (95%CI:13.0–25.4), whereas infants born of mothers who did not use iron supplements during pregnancy had a higher anemia prevalence at 66.7% (95%CI:30.0–148.8).

## Chi square analysis of factors associated with anaemia

Infants from Msambweni Sub-County had a significantly higher prevalence of anemia ($\chi^2$ = 17.0, df = 1, p<0.001) compared to infants from Matuga Sub-County.

Infant gender was significantly associated with anemia status ($\chi^2$ = 9.83, df = 1, p = 0.002). Infants iron deficiency status was significantly associated with anemia status ($\chi^2$ = 10.46, df = 1, p = 0.001).

Use of iron supplements during pregnancy was significantly associated with infants' anemia status ($\chi^2$ = 4.48, df = 1, p = 0.034) (Table 3).

## Bivariate analysis of factors associated with anaemia

Overall, infants in Msambweni Sub-County were 2.8 times, [OR = 2.80 (95%CI: 1.70–4.62, p<0.001)] more likely to be anemic compared to infants in Matuga Sub-County.

Male infants had significantly higher odds of anemia compared to female infants, OR = 2.20 (95%CI:1.33–3.63, p = 0.002). Similarly, iron deficient infants also had significantly higher odds of anemia than iron replete infants, OR = 2.35, (95%CI: 1.39–3.99, p = 0.001).

Mothers who did not use iron supplements during pregnancy had marginally significant higher odds of having infants with anemia than mothers who used iron supplements during pregnancy, OR = 9.0 (95%CI:0.79–102.8, p = 0.077) (Table 4).

## Multivariate analysis of factors associated with anaemia

Pairwise correlation of variables in the bivariate analysis was used to assess for correlation (Table 5). Stepwise variable selection with an inclusion criterion of p-value<0.5 was then done

**Table 2. Nutritional outcomes in infants.**

| Nutritional conditions | Wasting % (95% CI) | Underweight % (95% CI) | Stunting % (95%CI) | Iron deficient % (95%CI) | Anaemic % (95% CI) | Iron deficiency Anemia % (95%CI) |
|---|---|---|---|---|---|---|
| Overall | 2.1% (1.1–4.1); n = 9 | 0.2% (0.0–1.7); n = 1 | 6.2% (4.3–9.0); n = 26 | 60.4% (55.9–65.2); n = 256 | 21.0% (17.5–25.2); n = 89 | 15.8% (12.7–19.7); n = 67 |
| SubCounty | | | | | | |
| Kwale | 3.3% (1.6–6.9); n = 7 | 0.5% (0.1–3.4); n = 1 | 4.3% (2.3–8.1); n = 9 | 55.1% (49.1–62.6); n = 117 | 12.8% (9.0–18.2); n = 27 | 9.5% (6.2–14.4); n = 20 |
| Msambweni | 1.0% (0.2–3.8); n = 2 | 0% | 8.1% (5.2–12.8); n = 17 | 65.3% (59.2–72.0); n = 139 | 29.1% (23.6–35.9); n = 62 | 22.1% (17.1–28.4); n = 47 |
| Location: | | | | | | |
| Mkongani | 2.4% (0.6–9.6); n = 2 | 0% | 4.9% (1.9–12.7); n = 4 | 59.6% (50.0–71.4); n = 49 | 14.6% (8.7–24.7); n = 12 | 9.8% (5.1–18.8); n = 8 |
| Tsimba Golini | 4.3% (1.8–10.1); n = 5 | 0.9% (0.1–6.0); n = 1 | 4.3% (1.8–10.1); n = 5 | 50.4% (42.1–60.4); n = 59 | 12.8% (8.0–20.6); n = 15 | 10.3% (6.0–17.5); n = 12 |
| Waa | 0% | 0% | 0% | 75.0% (51.1–104); n = 9 | 0% | 0% |
| Kinondo | 0% | 0% | 0% | 61.5% (40.0–94.6); n = 8 | 50.8% (13.6–69.5); n = 4 | 15.4% (4.3–55); n = 2 |
| Ramisi | 2.7% (0.7–10.7); n = 2 | 0% | 11.0% (5.7–21.1); n = 8 | 66.2% (56.3–78); n = 49 | 27.0% (18.6–39.3) | 24.3% (16.3–36.4); n = 18 |
| Ukunda | 0% | 0% | 7.1% (3.5–14.6); n = 7 | 62.0% (53.2–72.3); n = 62 | 27.0% (19.6–37.3); n = 27 | 18.0% (11.8–27.3); n = 18 |
| Gombato | 0% | 0% | 7.7% (2.0–29.1); n = 2 | 77.0% (62.3–94.9); n = 20 | 42.3% (27.0–66.3); n = 11 | 34.6% (20.4–58.7); n = 9 |
| Age of the mother | | | | | | |
| ≤20 | 0% | 2.1% (0.3–14.8); n = 1 | 6.4% (2.1–19.1); n = 3 | 72.3% (60.6–86.3); n = 34 | 19.1% (10.6–34.5); n = 9 | 17.0% (9.1–32.0); n = 8 |
| 21–30 | 2.1% (0.9–4.9); n = 5 | 0% | 5.8% (3.5–9.6); n = 14 | 61.9% (56.1–68.3); n = 151 | 20.5% (16.0–26.2); n = 50 | 14.8% (10.9–19.-9); n = 36 |
| 31–40 | 3.4% (1.3–9.0); n = 4 | 0% | 6.0% (2.9–12.3); n = 7 | 52.1% (43.9–61.9); n = 62 | 24.4% (17.8–33.4); n = 29 | 18.5% (12.7–27.0); n = 22 |
| >40 | 0% | 0% | 16.7% (4.7–59.1); n = 2 | 75.0% (54.104); n = 9 | 8.3% (1.3–54.4); n = 1 | 8.3% (1.3–54.4); n = 1 |
| Mother's level of education | | | | | | |
| None | 0% | 0% | 0% | 60.0% (36.2–99.5); n = 6 | 10.0% (1.6–64.2); n = 1 | 10.0% (1.6–64.2); n = 1 |
| Primary | 2.4% (0.6–9.3); n = 2 | 0% | 3.5% (1.2–10.7); n = 3 | 62.4% (52.9–73.6); n = 53 | 22.4% (15.0–33.2); n = 19 | 14.1% (8.4–23.8); n = 12 |
| Secondary | 0% | 0% | 0% | 50% (35.7–70); n = 17 | 11.8% (4.7–29.5); n = 4 | 8.8% (3.0–26.0); n = 3 |
| Post-secondary | 11.5% (4.0–33.4); n = 3 | 0% | 15.5% (6.2–37.9); n = 4 | 64.3% (48.8–84.7); n = 18 | 21.4% (10.5–43.6); n = 6 | 17.9% (8.1–39.5); n = 5 |
| Parity | | | | | | |
| 1 | 2.2% (0.6–8.7); n = 2 | 1.1% (0.2–7.7); n = 1 | 8.8% (4.5–17); n = 8 | 61.3% (52.1–72.0); n = 57 | 23.7% (16.4–34.1); n = 22 | 18.3% (11.9–28.1); n = 17 |
| 2 | 2.7% (0.9–8.2); n = 3 | 0% | 3.6% (1.4–9.3); n = 4 | 60.7% (52.3–70.5); n = 68 | 18.8% (12.8–27.6); n = 21 | 12.5% (7.7–20.4); n = 14 |
| 3 | 1.1% (0.2–7.9); n = 1 | 0% | 5.6% (2.4–13.2); n = 5 | 62.9% (53.6–73.8); n = 56 | 21.3% (14.3–31.8); n = 19 | 19.1% (12.5–29.3); n = 17 |
| ≥4 | 2.4% (0.8–7.4); n = 3 | 0% | 5.6% (2.7–11.6); n = 7 | 60.0% (52.-69.2); n = 75 | 21.6% (15.5–30.2); n = 27 | 15.2% (10.0–23.0); n = 19 |
| Mother anemic | | | | | | |

(*Continued*)

**Table 2.** (Continued)

| Nutritional conditions | Wasting % (95% CI) | Underweight % (95% CI) | Stunting % (95%CI) | Iron deficient % (95%CI) | Anaemic % (95% CI) | Iron deficiency Anemia % (95%CI) |
|---|---|---|---|---|---|---|
| Yes | 2.9% (1.2–7.0); n = 5 | 0.6% (0.1–4.2%); n = 1 | 6.5% (3.7–11.5); n = 11 | 63.0% (56.2–70.6); n = 109 | 22.0% (16.6–29.1); n = 38 | 17.9% (13.0–24.7); n = 31 |
| No | 1.2% (0.3–4.9); n = 2 | 0% | 6.8% (3.9–12.1); n = 11 | 67.1% (60.2–74.7); n = 108 | 21.7% (16.2–29.1); n = 35 | 16.2% (11.4–23.0); n = 26 |
| **Weight of the mother** | | | | | | |
| <50 | 1.0% (0.1–7.2); n = 1 | 0% | 5.1% (2.2–12.0); n = 5 | 61.2% (52.3–71.7); n = 60 | 17.3% (11.3–26.7); n = 17 | 14.3% (8.8–23.2); n = 14 |
| 50–59 | 0.7% (0.1–4.9); n = 1 | 0% | 4.9% (2.4–10.1); n = 7 | 58.2% (50.7–66.8); n = 85 | 26% (19.8–34.2); n = 38 | 19.2% (13.8–26.8); n = 28 |
| 60–69 | 3.1% (1.0–9.3); n = 3 | 1% (0.1–7.2); n = 1 | 8.2% (4.2–15.9); n = 8 | 56.1% (47.1–66.9); n = 55 | 20.4% (13.8–30.2); n = 20 | 16.3% (10.4–25.6); n = 16 |
| 70–79 | 4.3% (1.1–16.5); n = 2 | 0% | 6.4% (2.1–19.1); n = 3 | 68.1% (56.0–82.8); n = 32 | 25.5% (15.7–41.6); n = 12 | 14.9% (7.5–29.5); n = 7 |
| >80 | 5.9% (1.5–22.6); n = 2 | 0% | 8.8% (3.0–26); n = 3 | 70.6% (56.8–87.8); n = 24 | 5.9% (1.5–22.6); n = 2 | 5.9% (1.5–22.6); n = 2 |
| **Mother uses iron supplement** | | | | | | |
| Yes | 3.3% (1.3–7.8); n = 5 | 0% | 4.6% (2.2–9.5); n = 7 | 60.4% (53.1–68.6); n = 93 | 18.2% (13.0–25.4); n = 28 | 13.6% (9.2–20.3); n = 21 |
| No | 0% | 0% | 0% | 33.3% (6.7–165.1); n = 1 | 66.7% (30.0–148.4); n = 2 | 0% |
| **Infant gender** | | | | | | |
| Male | 3.9% (2.1–7.5); n = 9 | 0.4% (0.1–3.1); n = 1 | 5.7% (3.3–9.6); n = 13 | 64.2% (58.3–70.7); n = 149 | 26.7% (21.6–33.1); n = 62 | 20.3% (15.7–26.2); n = 47 |
| Female | 0% | 0% | 6.8% (4.0–11.6); n = 13 | 56.3% (49.7–63.8); n = 107 | 14.2% (10.0–20.2); n = 27 | 10.5% (7.0–15.9); n = 20 |
| **Infant weight at birth** | | | | | | |
| <2.5 | 3.2% (0.5–22.2); n = 1 | 0% | 32.3% (19.4–53.7); n = 10 | 71% (56.7–88.9); n = 22 | 29% (16.7–50.3); n = 9 | 22.3% (11.8–43.3); n = 7 |
| ≥2.5 | 2.1% (1.0–4.1); n = 8 | 0.3% (0.0–1.8); n = 1 | 4.2% (2.6–6.7); n = 16 | 59.4% (54.7–64.5); n = 230 | 20.2% (16.5–24.6); n = 78 | 15% (11.8–19.0); n = 58 |
| **Infant weight at 6 weeks** | | | | | | |
| <5.0 | 3.0% (1.6–5.8); n = 9 | 0.3% (0.0–2.4); n = 1 | 8.8% (6.1–12.7); n = 26 | 59.3% (53.9–65.1); n = 176 | 19.9% (15.8–25); n = 59 | 16.2% (12.5–20.9); n = 48 |
| ≥5.0 | 0% | 0% | 0% | 63.5% (55.6–72.5); n = 80 | 23.8% (17.4–32.5); n = 30 | 15.1% (1–22.8); n = 19 |
| **Gestational weeks** | | | | | | |
| <37 | 2.0% (0.3–13.6); n = 1 | 0% | 2.0% (0.3–13.7); n = 1 | 60.4% (48.5–75.1); n = 32 | 24.5% (15.3–39.3); n = 13 | 18.9% (10.8–33.0); n = 10 |
| ≥37 | 2.2% (1.1–4.3); n = 8 | 0.3% (0.0–1.9); n = 1 | 6.6% (4.5–9.7); n = 24 | 60.9% (56.1–66.1); n = 223 | 20.5% (16.7–25.1); n = 75 | 15.6% (12.3–19.8); n = 57 |
| **Infant's height (cm)** | | | | | | |
| <55 | 0.5% (0.1–3.5); n = 1 | 0.5% (0.1–3.5); n = 1 | 13.1% (9.1–18.7); n = 26 | 59.5% (53.1–66.7); n = 119 | 22.0% (16.9–28.6); n = 44 | 17.0% (12.5–23.1); n = 34 |
| ≥55 | 3.6% (1.8–7.1); n = 8 | 0% | 0% | 61.4% (55.4–68.2); n = 137 | 20.2% (15.5–26.2); n = 45 | 14.8% (10.8–20.3); n = 33 |
| **Infant's head circumference** | | | | | | |
| <38 | 2.2% (1.0–4.9); n = 6 | 0.4% (0.1–2.6); n = 1 | 9.3% (6.4–13.5); n = 25 | 63.5% (58.0–69.5); n = 172 | 20.3% (16.0–25.7); n = 55 | 15.9% (12.1–20.9); n = 43 |
| ≥38 | 2.0% (0.6–6.1); n = 3 | 0% | 0.7% (0.1–4.7); n = 1 | 55.3% (47.9–63.8); n = 84 | 22.4% (16.6–30.1); n = 34 | 15.8% (10.9–22.8); n = 24 |

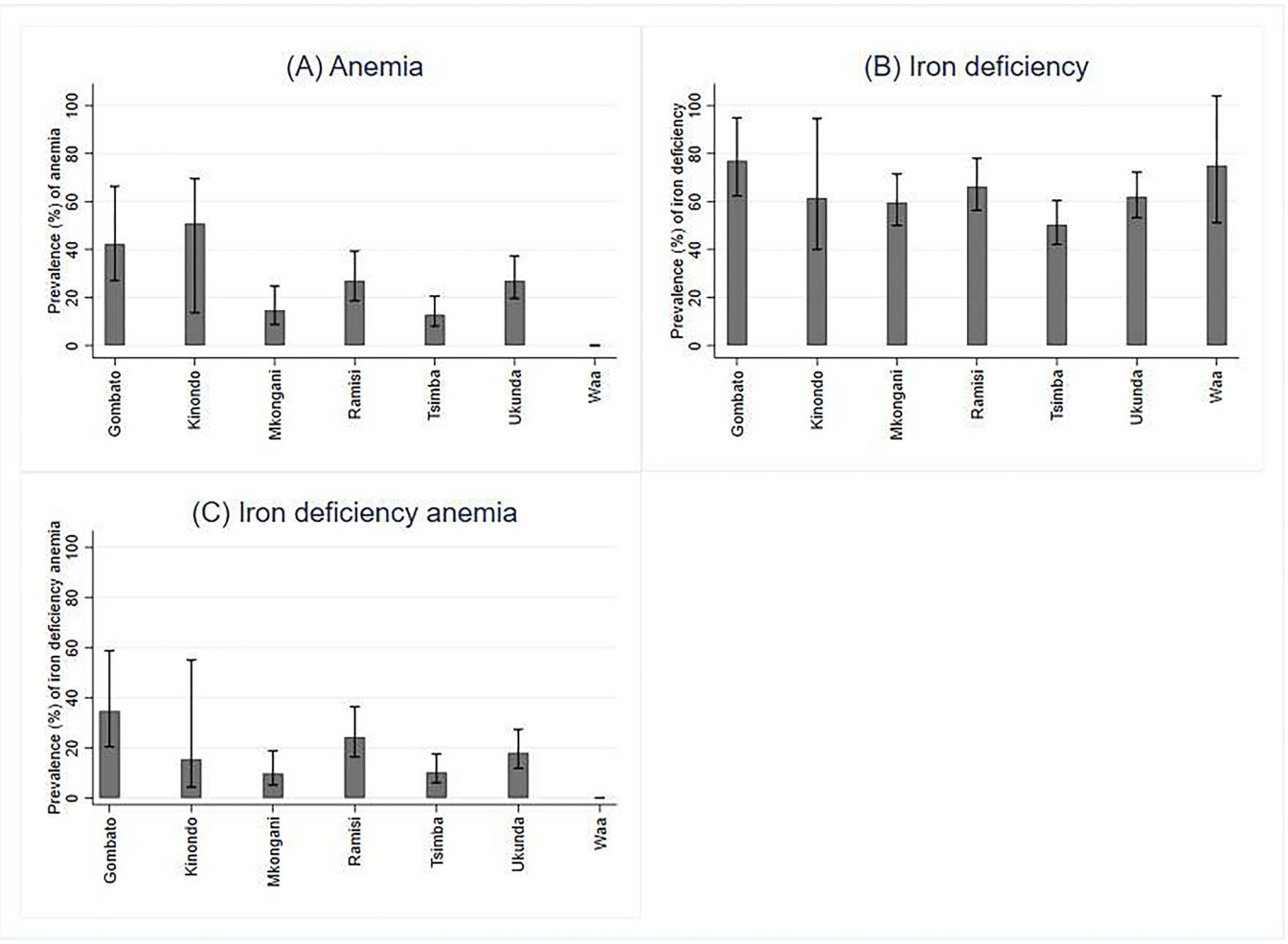

**Fig 1. Prevalence of ID, anemia and IDA among six-week-old infants by locations in Kwale County, Kenya.**

which included variables such as parity, birthweight, gender, age of the mother, use of iron supplements, education level and iron deficiency in the multivariate analysis model.

Significantly higher odds of anemia were seen infants born of mothers who did not use iron supplements during pregnancy, aOR = 74.01 (95%CI: 2.45–2238.21, p = 0.013). Mothers with parity $\geq$ 4 had significantly lower odds of having infants with anemia aOR = 0.05 (95% CI:0.00–0.77, p = 0.024) (Table 6).

## Discussion

This study among six-week-old infants in south Kenya found the prevalence of anemia, ID and IDA to be 21.0%, 60.4% and 15.8% respectively. Significant factors associated with anemia were infant iron deficiency status, gender, maternal use of iron supplements during pregnancy and the sub-County of residence. On multivariate analysis, factors that remained significant were use of iron supplements during pregnancy and parity of $\geq$4.

Anemia prevalence in this study is slightly lower than the global anemia prevalence of 22.8% across all ages [24] and 42% in children < 5 years of age [3]. Similarly, anemia prevalence in this study is lower than the prevalence in older Kenyan infants where 70% of infants at 6-months of age are anemic [5] and the nationwide anemia prevalence in Kenyan children

**Table 3. Chi-square association of anaemia with selected predictors among six-week-old infants in Kwale County, Kenya.**

| Selected predictors | Iron deficient | | Anaemic | | Iron deficiency anemia | |
|---|---|---|---|---|---|---|
| | Yes | No | Yes | No | Yes | No |
| SubCounty | | | | | | |
| Kwale | 117 | 94 | 27 | 184 | 20 | 191 |
| Msambweni | 139 | 74 | 62 | 151 | 47 | 166 |
| $\chi^2$(test statistic, p-value) | **$\chi^2$ = 4.26, df = 1, p = 0.039*** | | **$\chi^2$ = 17.00, df = 1, p = 0.000*** | | **$\chi^2$ = 12.62, df = 1, p = 0.000*** | |
| Age of the mother | | | | | | |
| ≤20 | 34 | 13 | 9 | 38 | 8 | 39 |
| 21–30 | 151 | 93 | 50 | 194 | 36 | 208 |
| 31–40 | 62 | 57 | 29 | 90 | 22 | 97 |
| >40 | 9 | 3 | 1 | 11 | 1 | 11 |
| $\chi^2$(test statistic, p-value) | $\chi^2$ = 7.53, df = 3, p = 0.057 | | $\chi^2$ = 2.10, df = 3, p = 0.552 | | $\chi^2$ = 1.40, df = 3, p = 0.707 | |
| Mother's level of education | | | | | | |
| None | 6 | 4 | 1 | 9 | 1 | 9 |
| Primary | 53 | 32 | 19 | 66 | 12 | 73 |
| Secondary | 17 | 17 | 4 | 30 | 3 | 31 |
| Post-secondary | 18 | 10 | 6 | 22 | 5 | 23 |
| $\chi^2$(test statistic, p-value) | $\chi^2$ = 1.82, df = 3, p = 0.610 | | $\chi^2$ = 2.40, df = 3, p = 0.494 | | $\chi^2$ = 1.23, df = 3, p = 0.745 | |
| Parity | | | | | | |
| 1 | 57 | 36 | 22 | 71 | 17 | 76 |
| 2 | 68 | 44 | 21 | 91 | 14 | 98 |
| 3 | 56 | 33 | 19 | 70 | 17 | 72 |
| ≥4 | 75 | 40 | 27 | 98 | 19 | 106 |
| $\chi^2$(test statistic, p-value) | $\chi^2$ = 0.20, df = 3, p = 0.978 | | $\chi^2$ = 0.75, df = 3, p = 0.861 | | $\chi^2$ = 2.08, df = 3, p = 0.556 | |
| Mother anemic | | | | | | |
| Yes | 109 | 64 | 38 | 135 | 31 | 142 |
| No | 108 | 53 | 35 | 126 | 26 | 135 |
| $\chi^2$(test statistic, p-value) | $\chi^2$ = 0.61, df = 1, p = 0.435 | | $\chi^2$ = 0.0025, df = 1, p = 0.960 | | $\chi^2$ = 0.18, df = 1, p = 0.667 | |
| Mother uses iron supplement | | | | | | |
| Yes | 93 | 61 | 28 | 126 | 21 | 133 |
| No | 1 | 2 | 2 | 1 | 0 | 3 |
| $\chi^2$(test statistic, p-value) | $\chi^2$ = 0.90, df = 1, p = 0.344 | | **$\chi^2$ = 4.48, df = 1, p = 0.034*** | | $\chi^2$ = 0.47, df = 1, p = 0.492 | |
| Infant gender | | | | | | |
| Male | 149 | 83 | 62 | 170 | 47 | 185 |
| Female | 107 | 83 | 27 | 163 | 20 | 170 |
| $\chi^2$(test statistic, p-value) | $\chi^2$ = 2.74, df = 1, p = 0.098 | | **$\chi^2$ = 9.83, df = 1, p = 0.002*** | | **$\chi^2$ = 7.41, df = 1, p = 0.006*** | |
| Infant weight at birth | | | | | | |
| <2.5 | 22 | 9 | 9 | 22 | 7 | 24 |
| ≥2.5 | 230 | 157 | 78 | 309 | 58 | 329 |
| $\chi^2$(test statistic, p-value) | $\chi^2$ = 1.60, df = 1, p = 0.207 | | $\chi^2$ = 1.37, df = 1, p = 0.241 | | $\chi^2$ = 1.26, df = 1, p = 0.262 | |
| Gestational weeks | | | | | | |
| <37 | 32 | 21 | 13 | 40 | 10 | 43 |
| ≥37 | 223 | 143 | 75 | 291 | 57 | 309 |
| $\chi^2$(test statistic, p-value) | $\chi^2$ = 0.01, df = 1, p = 0.939 | | $\chi^2$ = 0.45, df = 1, p = 0.500 | | $\chi^2$ = 0.37, df = 1, p = 0.541 | |
| Iron deficient | | | | | | |
| Yes | Not applicable | Not applicable | 67 | 189 | 67 | 189 |
| No | Not applicable | Not applicable | 22 | 146 | 0 | 168 |
| $\chi^2$(test statistic, p-value) | Not applicable | | **$\chi^2$ = 10.46, df = 1, p = 0.001*** | | **$\chi^2$ = 52.22, df = 1, p = 0.000*** | |

**Table 4. Bivariate logistic regression analysis of factors associated with anemia among six-week-old infants in Kwale County, Kenya.**

| Factors | N (%) | Odds Ratio (95% CI) | P-value |
|---|---|---|---|
| Infant's factors: | | | |
| Gestation weeks | | | |
| <37 weeks | | 1.26 (0.64–2.48) | 0.501 |
| ≥37 weeks | | Reference | |
| Gender | | | |
| Male | | **2.20 (1.33–3.63)** | **0.002***  |
| Female | | Reference | |
| Birth weight | | | |
| <2.5kg | | 1.62 (0.72–3.66) | 0.245 |
| ≥2.5kg | | Reference | |
| Wasted | | | |
| < 2.0 SD | 9 (2.1%) | Insufficient observations | - |
| ≥2.0 SD | 411 (97.9%) | Reference | |
| Underweight | | | |
| <2.0 SD | 1 (0.2%) | Insufficient observations | - |
| ≥2.0SD | 419 (99.8%) | Reference | |
| Stunted | | | |
| < 2.0 SD | 26 (6.2%) | 1.16 (0.45–2.98) | 0.759 |
| ≥2.0 SD | 394 (93.8%) | Reference | |
| Weight at 6 weeks | | | |
| <5.0kg | | 0.79 (0.48–1.31) | 0.363 |
| ≥5.0kg SD | | Reference | |
| Height at 6 weeks | | | |
| <55 | | 1.12 (0.70–1.78) | 0.647 |
| ≥55 | | Reference | |
| Head circumference | | | |
| <38 | | 0.88 (0.56–1.43) | 0.616 |
| ≥38 | | Reference | |
| Iron deficient | | | |
| Yes | | **2.35 (1.39–3.99)** | **0.001***  |
| No | | Reference | |
| Maternal factors: | | | |
| Anemic | | | |
| Yes | | 1.01 (0.60–1.70) | 0.960 |
| No | | Reference | |
| Use of iron supplements | | | |
| Yes | | Reference | |
| No | | 9.0 (0.79–102.8) | 0.077 |
| Age | | | |
| ≤20 | | Reference | |
| 21–30 | | 1.09 (0.49–2.40) | 0.834 |
| 31–40 | | 1.36 (0.59–3.15) | 0.472 |
| >40 | | 0.38 (0.04–3.37) | 0.388 |
| Weight | | | |
| <50 | | Reference | |
| 50–59 | | 1.68 (0.88–3.18) | 0.114 |
| 60–69 | | 1.22 (0.60–2.50) | 0.584 |

(*Continued*)

**Table 4.** (Continued)

| Factors | N (%) | Odds Ratio (95% CI) | P-value |
|---|---|---|---|
| 70–79 | | 1.63 (0.71–3.78) | 0.251 |
| ≥80 | | 0.30 (0.07–1.36) | 0.119 |
| Parity | | | |
| 1 | | Reference | |
| 2 | | 0.74 (0.4–1.5) | 0.391 |
| 3 | | 0.88 (0.4–1.8) | 0.710 |
| ≥4 | | 0.89 (0.47–1.69) | 0.719 |
| Education | | | |
| None | | Reference | |
| Primary | | 2.59 (0.31–25.8) | 0.381 |
| Secondary | | 1.19 (0.12–12.14) | 0.877 |
| Post-secondary | | 2.45 (0.26–25.39) | 0.435 |
| Sub-County: | | | |
| Kwale | | Reference | |
| Msambweni | | **2.80 (1.70–4.62)** | **<0.001\*** |

*Indicates statistically significant factors (p<0.05)

under one year of 39.5% [13]. In addition, this prevalence is also lower compared to a prevalence of 69% in a recent non-representative (including infants in Msambweni Sub-County only) study in the same area of Kenya among infants aged 6–10 months [24]. This is not surprising as younger infants generally have high hemoglobin (Hb) levels in the first few weeks

**Table 5. Pairwise correlation coefficients of factors associated with anemia among six-week-old infants in Kwale County, Kenya.**

| | ANM | GES | PAR | BW | GEN | IDA | WAS | UWT | STU | AMO | UIS | AGM | WOM | EDS | WOC | HOC | HCC |
|---|---|---|---|---|---|---|---|---|---|---|---|---|---|---|---|---|---|
| ANM | 1.0000 | | | | | | | | | | | | | | | | |
| GES | 0.0329 | 1.0000 | | | | | | | | | | | | | | | |
| PAR | -0.0329 | -0.0571 | 1.0000 | | | | | | | | | | | | | | |
| BW | 0.0573 | -0.0234 | 0.0068 | 1.0000 | | | | | | | | | | | | | |
| GEN | -0.1526 | 0.0602 | -0.0360 | 0.1107 | 1.0000 | | | | | | | | | | | | |
| IDA | 0.1571 | -0.0038 | -0.0145 | 0.0618 | -0.0805 | 1.0000 | | | | | | | | | | | |
| WAS | -0.0756 | -0.0052 | 0.0114 | 0.0206 | -0.1350 | -0.0493 | 1.0000 | | | | | | | | | | |
| UWT | 0.0956 | -0.0183 | -0.0624 | -0.0140 | -0.0446 | 0.0393 | -0.0072 | 1.0000 | | | | | | | | | |
| STU | 0.0150 | -0.0637 | -0.0276 | 0.3048 | 0.0240 | 0.0448 | 0.0302 | 0.1902 | 1.0000 | | | | | | | | |
| AMO | 0.0027 | -0.0582 | 0.0449 | 0.0552 | 0.0148 | -0.0427 | 0.0590 | 0.0536 | -0.0073 | 1.0000 | | | | | | | |
| UIS | 0.1688 | -0.0571 | 0.1548 | -0.0404 | -0.1347 | -0.0756 | -0.0256 | . | -0.0306 | -0.1686 | 1.0000 | | | | | | |
| AGM | 0.0151 | -0.0536 | 0.6186 | -0.0289 | 0.0064 | -0.0880 | 0.0483 | -0.0887 | 0.0318 | 0.0245 | 0.0054 | 1.0000 | | | | | |
| WOM | -0.0401 | -0.0040 | 0.0803 | -0.1243 | 0.0573 | 0.0503 | 0.1074 | 0.0218 | 0.0486 | -0.1651 | -0.0279 | 0.1856 | 1.0000 | | | | |
| EDS | -0.0054 | 0.1213 | -0.3322 | 0.0237 | 0.0377 | -0.0136 | 0.1533 | . | 0.1679 | -0.1685 | -0.0287 | 0.0161 | 0.2331 | 1.0000 | | | |
| WOC | -0.0443 | 0.0460 | 0.0334 | 0.1427 | 0.1741 | -0.0396 | 0.0958 | 0.0316 | 0.1663 | 0.0824 | -0.1319 | -0.0486 | -0.1808 | -0.0788 | 1.0000 | | |
| HOC | -0.0223 | -0.0318 | -0.0289 | -0.2055 | -0.1808 | 0.0198 | 0.1075 | -0.0515 | -0.2707 | -0.0393 | -0.0583 | 0.0511 | 0.1698 | 0.0699 | -0.4408 | 1.0000 | |
| HCC | 0.0244 | -0.0930 | -0.0441 | -0.1344 | -0.1532 | -0.0805 | -0.0081 | -0.0366 | -0.1719 | -0.0333 | 0.0124 | 0.0414 | 0.0810 | -0.0398 | -0.2771 | 0.3244 | 1.0000 |

ANM: Anemia, GES: Gestational weeks, PAR: Parity, BW: Birth weight, GEN: Gender, IDA: Iron deficient anemia, WAS: Wasting, UWT: Underweight, STU: Stunted, AMO: Anemic mother, UIS: Use of iron supplements, AGM: Age of the mother, WOM: Weight of the mother, EDS: Education level, WOC: Weight of the child, HOC: Height of the child, HCC: Head circumference of the child.

**Table 6. Multivariate logistic regression analysis of factors associated with anemia among six-week-old infants in Kwale County, Kenya.**

| Factors | Adjusted Odds Ratio (95% CI) | P-Value |
|---|---|---|
| Infant's factors: | | |
| Gestation weeks | | |
| <37 weeks | 0.65 (0.15–2.82) | 0.562 |
| ≥37 weeks | Reference | |
| Gender | | |
| Male | 1.92 (0.66–5.54) | 0.229 |
| Female | Reference | |
| Birth weight | | |
| <2.5kg | 1.28 (0.20–8.25) | 0.798 |
| ≥2.5kg | Reference | |
| Stunted | | |
| < 2.0 SD | 0.20 (0.01–3.86) | 0.285 |
| ≥2.0 SD | Reference | |
| Weight at 6 weeks | | |
| <5.0kg | 1.14 (0.31–4.19) | 0.844 |
| ≥5.0kg | Reference | |
| Height at 6 weeks | | |
| <55 | 1.31 (0.41–4.22) | 0.652 |
| ≥55 | Reference | |
| Head circumference | | |
| <38 | 0.76 (0.25–2.36) | 0.639 |
| ≥38 | Reference | |
| Iron deficient | | |
| Yes | 2.15(0.68–6.81) | 0.192 |
| No | Reference | |
| Maternal factors: | | |
| Use of iron supplements | | |
| Yes | Reference | |
| No | **74.01(2.45–2238.21)** | **0.013**[*] |
| Age | | |
| ≤20 | Reference | |
| 21–30 | 1.65(0.19–14.36) | 0.649 |
| 31–40 | 11.37(0.69–186.86) | 0.089 |
| >40 | | |
| Weight | | |
| <50 | Reference | |
| 50–59 | 1.80(0.39–8.43) | 0.454 |
| 60–69 | 1.81(0.37–8.88) | 0.463 |
| 70–79 | 1.05(0.16–6.93) | 0.963 |
| ≥80 | 0.24(0.01–4.14) | 0.330 |
| Parity | | |
| 1 | Reference | |
| 2 | 0.52(0.09–3.03) | 0.464 |
| 3 | 0.38(0.05–2.57) | 0.318 |
| ≥4 | **0.06(0.00–0.69)** | **0.024**[*] |
| Education | | |

*(Continued)*

**Table 6.** (Continued)

| Factors | Adjusted Odds Ratio (95% CI) | P-Value |
| --- | --- | --- |
| None | Reference | |
| Primary | 3.10(0.28–34.02) | 0.354 |
| Secondary | 0.53(0.04–7.88) | 0.646 |
| Post-secondary | 2.82(0.17–47.02) | 0.470 |
| Subcounty: | | |
| Kwale | Reference | |
| Msambweni | 2.28(0.78–6.62) | 0.129 |

*Indicates statistically significant factors (p<0.005)

after birth due to high levels of circulating fetal hemoglobin (HbF); these levels decrease progressively over the first 6-months as HbF is replaced by adult hemoglobin (HbA) [25]. According to WHO categorization of anemia (severe; ≥40%, moderate; 20.0%-39.9%, mild; 5.0%-19.9% and normal ≤4.9%), the 21.0% anemia prevalence is of moderate public health significance [26].

Prevalence of ID and IDA in this study was 60.4% and 15.8% respectively. This means ID accounted for 75.3% of the total anemia cases in this study. This agrees with documented global anemia burden where ID accounted for ≥60% of the cases of anemia in children <5 years in LMIC who predominantly have the highest burden of anemia [2].

Infant gender significantly predicted anemia status with male infants having an increased risk of anemia compared to female infants. Few studies have been done to address the question of gender differences on anemia and iron status in early infancy. Some studies have reported higher anemia prevalence in boys compared to girls [5, 27]. These differences may reflect increased cases of ID in boys and a potential contribution of genetics [28]. In one study anemia in infancy was associated with alterations in systemic metabolism dependent on gender with male infants having higher prevalence of anemia and ID due to greater oxidative stress and microbial dysfunction compared to female infants [29].

In this study, the use of iron supplementation emerged as a significant predictor of infants' anemia status. Infants born to mothers who did not report to using iron supplements during pregnancy exhibited an elevated risk of anemia compared to infants born to mothers who reported using iron supplements during pregnancy. This is in agreement with other clinical trials studies on effect of daily iron supplementation in conjunction with folic acid and other vitamins and minerals for pregnant women as an intervention in antenatal care, which showed that supplementation with iron reduced the risk of ID and maternal anemia [30]. During pregnancy, iron supplementation produces more benefits to both the mother and the baby by promoting fetal health, preparing the fetus with adequate iron stores to be used during early postnatal stages of life as well as reducing maternal morbidity [32].

In this study, infants in Msambweni Sub-County had significantly higher risk of anemia compared to infants in Matuga Sub-County. While there was no direct measure of diet or other factors leading to this, evidence shows that low diet quality, inadequate dietary diversity and consumption of monotonous diets in Africa among poor populations are important predictors of maternal anemia and poor birth outcomes [31]. The 2017 Kwale County Maternal Infant and Young Child Nutrition (MIYCN) Knowledge, Attitude and Practices (KAP) survey report states that the minimum dietary diversity for women of reproductive age is 45.2%, suggesting a low-quality diet [32]. The report notes that factors influencing maternal nutrition were consistent across various sub-Counties. The primary reasons identified included limited

purchasing power to acquire adequate food, food insecurity and cultural factors impacting the diet choices of certain women during pregnancy in specific communities [32]. A meta-analysis on dietary diversity among pregnant and lactating women in Ethiopia showed that only 41% of pregnant women and 50% of lactating women had adequate dietary diversity with factors leading to this including wealth index level of household, maternal education and nutritional information [33]. These findings suggest that there is need to improve maternal and child nutrition services particularly on nutrition education on the selection and diversification of meals.

In this study, infants of mothers with parity of four or more were significantly less likely to be anemic compared to infants born of mothers with parity of one. This is in contrast to other studies where higher maternal parity was associated with greater risk for childhood anemia [27]. However, these studies examined older infants and having many children in the family may reduce the ability to feed them appropriately. The contrast in our study could be because mothers who have more than one child are more likely to comply and adhere to the guidelines and instructions on daily usage of iron and folic acid supplements given during pregnancy through education and sensitization in previous pregnancies from health experts.

Whilst our study did not show any significant association between gestational age, birth weight and maternal anemia with infants' anemia status, other studies have shown significant associations. Preterm and low birthweight infants have a higher risk of developing ID and IDA because of higher iron requirements compared to term and normal birth weight infants due to increased post-natal growth rate [34]. A study conducted in Bangladesh found a significant association between maternal anemia and occurrence of childhood anemia; the prevalence of anemia among children born to anemic mothers was notably higher at (62%) compared to those born to non-anemic mothers [35].

## Study strengths and limitations

One significant strength of this study lies in the recruitment of infants from villages surrounding hospitals, maternity wards, well-baby clinics and vaccination clinics. This approach enhances the likelihood of a more representative sample from the study area population. Furthermore, besides gathering information related to infants, maternal data was also collected to assess its association with infants' anemia status.

However, it is important to acknowledge several limitations in this study. Firstly, being nested within a parent study, it exclusively included infants meeting the inclusion criteria for that specific study. The reliance on mother-child health booklets for maternal information posed a limitation, as these booklets were unavailable for some mothers. Additionally, the accuracy of information regarding use of iron supplements during pregnancy depended on mothers' ability to recall and honestly disclose the information. This study also lacked data on mother' blood samples and dietary information. Notably, participants were not recruited at the household level but were mobilized to a central recruitment point within the County such as hospitals, well-baby clinics and maternity wards.

## Conclusion

Our study findings indicate that in rural Kenya, anemia in early infancy affects 21% of infants, with iron deficiency accounting for 75.3% of these anemia cases. Furthermore, the study highlights the use maternal iron supplements as a crucial factor in reducing the risk of infants developing anemia. Addressing the issue of iron deficiency during early childhood is essential to enhance hemoglobin levels in early infancy, and in the long term prevent physical and cognitive impairments associated with iron deficiency during this crucial developmental stage. It

may be advisable to revisit the current recommendations that target screening and provision of iron supplements only to infants aged 6–24 months given the, given the high prevalence of iron deficiency in this study.

## Supporting information

**S1 Data. Minimal anonymized dataset of six-week-old infants (infant-mother pairs) surveyed in Kwale County, Kenya.**
(XLSX)

## Author Contributions

**Conceptualization:** Joyce Mwakishalua, Simon Karanja, Raphael Lihana, Nicole Stoffel, Michael Zimmermann.

**Data curation:** Joyce Mwakishalua.

**Formal analysis:** Joyce Mwakishalua, Collins Okoyo.

**Investigation:** Joyce Mwakishalua, Simon Karanja, Raphael Lihana, Collins Okoyo, Nicole Stoffel, Michael Zimmermann.

**Methodology:** Joyce Mwakishalua, Simon Karanja, Raphael Lihana, Collins Okoyo, Nicole Stoffel, Michael Zimmermann.

**Project administration:** Joyce Mwakishalua.

**Software:** Collins Okoyo.

**Supervision:** Joyce Mwakishalua, Simon Karanja, Raphael Lihana, Collins Okoyo, Nicole Stoffel, Michael Zimmermann.

**Validation:** Collins Okoyo.

**Visualization:** Joyce Mwakishalua, Collins Okoyo.

**Writing – original draft:** Joyce Mwakishalua, Simon Karanja, Raphael Lihana, Collins Okoyo, Nicole Stoffel, Michael Zimmermann.

**Writing – review & editing:** Joyce Mwakishalua, Simon Karanja, Raphael Lihana, Collins Okoyo, Nicole Stoffel, Michael Zimmermann.

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
