## [Decision Letter · Decision Letter 0]

18 Jan 2024

PGPH-D-23-01767

Prevalence and predictors of anemia among six-week-old infants in Kwale County, Kenya: A cross-sectional study

Dear Dr.Mwakishalua ,

Thank you for submitting your manuscript to PLOS Global Public Health. After careful consideration, we feel that it has merit but does not fully meet PLOS Global Public Health’s publication criteria as it currently stands. Therefore, we invite you to submit a revised version of the manuscript that addresses the points raised during the review process.

We look forward to receiving your revised manuscript.

Kind regards,

Aditi Apte, MD PhD

Academic Editor

Journal Requirements:

Additional Editor Comments (if provided):

Reviewers' comments:

Reviewer's Responses to Questions

**Comments to the Author**

1. Does this manuscript meet PLOS Global Public Health’s publication criteria? Is the manuscript technically sound, and do the data support the conclusions? The manuscript must describe methodologically and ethically rigorous research with conclusions that are appropriately drawn based on the data presented.

Reviewer #1: Yes

Reviewer #2: Partly

2. Has the statistical analysis been performed appropriately and rigorously?

Reviewer #1: Yes

Reviewer #2: I don't know

3. Have the authors made all data underlying the findings in their manuscript fully available (please refer to the Data Availability Statement at the start of the manuscript PDF file)?

Reviewer #1: Yes

Reviewer #2: No

4. Is the manuscript presented in an intelligible fashion and written in standard English?

Reviewer #1: Yes

Reviewer #2: Yes

5. Review Comments to the Author

Reviewer #1: Firstly, I would congratulate the authors for studying an important issue of public health relevance. There are very few treatment regimens and recommendations in the under 6 months age group, hence the authors have tried throwing light on the gaps in this area. The findings will be very valuable for policy and action.

The authors have done good job of extracting maximum information for this nested study from the parent study. However I feel that the paper would benefit from minor editing.

Minor edits need to be done as follows:

General comments: The references in the text need to mentioned by their number rather than the authors. This uniformity needs to be maintained throughout the manuscript. All the significant values in the tale need to be highlighted for easier reading.

Specific comments:

Introduction:

Line number 71, 72 the reference number needs to be mentioned rather than the author names, similarly at line number 101, 102, 125, 126

Materials and methods:

Line 137 reference of the RCT needs to be mentioned

Results:

It has been mentioned in the methods as well as the results that the Hemoglobin at the mother’s last ANC visit was recorded and that value is used in all the analysis. I would add more value if you could mention the average weeks at this visit or at what weeks this last ANC visit was.

This study is a nested study and there was no maternal blood sample but certainly if instead of just recorded values of Hemoglobin if there was a maternal blood sample available it would have added value to this study

Line 281, 282 you have mentioned that mothers weighing 50-59 kg had higher proportion of anaemic infants than mothers weighing >80kg so does that mean that the in between range of weight the relation was not significant? Need to clarify

Line 288,289 There is a difference between the 2 counties’, is there any specific reason which can be identified, is the diet different in these 2 counties? There is no direct measure of diet in this study which also is one of the drawbacks as diet plays a important role in anaemia. Some measure of diet or some general information collected through and Focus Group Discussion would have given clarity on whether there were any major differences in the diet.

Reviewer #2: The manuscript was decently written. However, there are places that can be improved before moving forward -

1) For statistical analysis, there are confidence intervals with very wide bounds which usually indicate problems with the models. The authors need to scrutinize them in revision.

2) For data availability, it seems the raw dataset used for the analysis has not been uploaded. To improve reproducibility, the authors can consider providing it.

Other comments, along with their corresponding sections highlighted, are in the uploaded attachment.

6. PLOS authors have the option to publish the peer review history of their article (what does this mean?). If published, this will include your full peer review and any attached files.

**Do you want your identity to be public for this peer review?** For information about this choice, including consent withdrawal, please see our Privacy Policy.

Reviewer #1: No

Reviewer #2: No

---

## [Editor Report · Decision Letter 1]

6 Mar 2024

Prevalence and predictors of anemia among six-week-old infants in Kwale County, Kenya: A cross-sectional study

PGPH-D-23-01767R1

Dear Dr Mwakishalua,

We are pleased to inform you that your manuscript 'Prevalence and predictors of anemia among six-week-old infants in Kwale County, Kenya: A cross-sectional study' has been provisionally accepted for publication in PLOS Global Public Health.

Best regards,

Aditi Apte, MD PhD

Academic Editor